# Characterisation of Mid-Gestation Amniotic Fluid Cytokine and Bacterial DNA Profiles in Relation to Pregnancy Outcome in a Small Australian Cohort

**DOI:** 10.3390/microorganisms11071698

**Published:** 2023-06-29

**Authors:** Lisa F. Stinson, Yey Berman, Shaofu Li, Jeffrey A. Keelan, Jan E. Dickinson, Dorota A. Doherty, John P. Newnham, Matthew S. Payne

**Affiliations:** 1School of Molecular Sciences, The University of Western Australia, Perth, WA 6009, Australia; lisa.stinson@uwa.edu.au; 2Division of Obstetrics and Gynaecology, The University of Western Australia, Perth, WA 6008, Australia; yeelah.berman@uwa.edu.au (Y.B.); shaofu.li@uwa.edu.au (S.L.); jan.dickinson@uwa.edu.au (J.E.D.); dorota.doherty@uwa.edu.au (D.A.D.); john.newnham@uwa.edu.au (J.P.N.); 3School of Biomedical Sciences, The University of Western Australia, Perth, WA 6009, Australia; jeff.keelan@uwa.edu.au

**Keywords:** PCR (polymerase chain reaction), bacteria, preterm birth, amniotic fluid, microbiome, cytokines, inflammation

## Abstract

A well-established association exists between intrauterine bacteria and preterm birth. This study aimed to explore this further through documenting bacterial and cytokine profiles in Australian mid-gestation amniotic fluid samples from preterm and term births. Samples were collected during amniocenteses. DNA was extracted and the full-length 16S rRNA gene was amplified and sequenced. Levels of the cytokines IL-1β, IL-6, IL-10, TNF-α and MCP-1 were determined using the Milliplex MAGPIX system. Bacterial DNA profiles were low in diversity and richness, with no significant differences observed between term and preterm samples. No differences in the relative abundance of individual OTUs between samples were identified. IL-1β and TNF-α levels were significantly higher in samples containing reads mapping to *Sphingomonas* sp.; however, this result should be interpreted with caution as similar reads were also identified in extraction controls. IL-6 levels were significantly increased in samples with reads that mapped to *Pelomonas* sp., whilst TNF-α levels were elevated in fluid samples from pregnancies that subsequently delivered preterm. Bacterial DNA unlikely to have originated from extraction controls was identified in 20/31 (64.5%) mid-gestation amniotic fluid samples. Bacterial DNA profiles, however, were not predictive of preterm birth, and although cytokine levels were elevated in the presence of certain genera, the biological relevance of this remains unknown.

## 1. Introduction

Globally, preterm birth is the leading cause of mortality and morbidity in children under 5 years of age [1]. In Australia, 8.7% of infants are born preterm [2]. Intra-amniotic infection and inflammation are well-characterised causes of preterm labour and delivery. However, mounting evidence suggests that a bacterial DNA signature may be present in amniotic fluid samples from healthy, full term deliveries [3,4,5,6,7,8,9] as early as mid-gestation [10]. It is therefore unclear the extent to which the presence of bacteria in amniotic fluid may be predictive of preterm birth. Previous studies of mid-trimester amniotic fluid samples have found no correlation between the presence of bacterial DNA and later birth outcomes [10]. Nevertheless, certain bacterial taxa, such as *Ureaplasma* sp., *Mycoplasma* sp. and *Fusobacterium nucleatum*, have been identified as causative agents of preterm birth [11,12,13,14,15]. Early identification of such pathogens may be useful in identifying high-risk pregnancies that may benefit from antimicrobial treatment.

Both infection-mediated and so-called “sterile” intra-amniotic inflammation can trigger preterm labour [16]. Pathogenic infection of the gestational tissues can elicit an innate immune response via activation of pattern recognition receptors, resulting in the release of inflammatory cytokines and chemokines. These stimulate production of prostaglandins and matrix metalloproteinases, resulting in uterine contractions and degradation of the foetal membranes. This kind of inflammation-driven preterm labour has also been reported in the absence of any observable infection [17,18]. Increased levels of inflammatory cytokines and chemokines, such as interleukin-1β (IL-1β), interleukin-6 (IL-6), interleukin-10 (IL-10), tumour necrosis factor-α (TNF-α) and monocyte chemoattractant protein 1 (MCP-1), may be detected in amniotic fluid samples prior to the onset of preterm labour, allowing early detection of at-risk pregnancies [19,20,21,22,23,24,25,26]. Indeed, a point-of-care test for elevated amniotic fluid IL-6 levels has been developed previously [27].

To date, limited studies have characterised bacterial DNA profiles of mid-trimester amniotic fluid samples [6,7,8,10]. Importantly, those that have did not include measurement of intra-amniotic inflammation. This is an important gap, as inflammation is the major pathway through which pathogenic infection elicits preterm labour. Combined bacterial and inflammatory analysis is therefore likely needed to differentiate term- and preterm birth-associated profiles when measured in mid-pregnancy. Here, we aimed to characterise markers of intra-amniotic inflammation and bacterial DNA profiles in mid-trimester amniotic fluid samples from a cohort of Australian women who went on to deliver preterm or at term.

## 2. Materials and Methods

### 2.1. Participant Recruitment

Women were recruited from King Edward Memorial Hospital (KEMH), Perth, Western Australia. Inclusion criteria were women with a singleton pregnancy referred for genetic amniocentesis to exclude aneuploidy based on high-risk findings at either first trimester screening (ultrasound measurement of foetal nuchal translucency and maternal biochemistry) or second trimester maternal serum screening (maternal biochemistry) with no apparent foetal structural anomalies. Gestational age (GA) was determined via ultrasound imaging and biometry in the first or early second trimester. Cases subsequently shown to be complicated by aneuploidy were withdrawn from the study. Approval for the study was obtained from the ethics committees of the Women and Newborn Health Service, Western Australia (1880/EW), and The University of Western Australia (RA/4/1/4784).

### 2.2. Sample Collection

All amniocenteses were performed between 15–20 weeks’ GA by medical specialists under ultrasound guidance. Three millilitres of amniotic fluid were aspirated in addition to the 10–15 mL collected for clinical measurement and subsequently dispensed into 1 mL aliquots in sterile cryovials prior to storage at −80 °C.

### 2.3. Pregnancy Outcome Data

Pregnancy outcome data were accessed via medical record review following completion of the pregnancy.

### 2.4. Cytokine Analyses

Concentrations of IL-1β, IL-6, IL-10, TNF-α and MCP-1 were determined via multiplex assay using the Milliplex MAGPIX system (EMD-Millipore/Merck KGaA, Darmstadt, Germany) with appropriate calibration curves as per Payne et al. [28].

### 2.5. DNA Extraction

DNA was extracted from amniotic fluid using a Mo Bio PowerMag Microbiome RNA/DNA isolation kit on an automated Kingfisher Duo platform. Briefly, 1 mL amniotic fluid samples were centrifuged at 40,000× *g* for 10 min @ 4 °C and all but 75 µL of supernatant was removed. Pellets were resuspended in 650 µL of lysis buffer/β-mercaptoethanol and then an additional 100 μL of phenol:chloroform:isoamyl alcohol (PCI) (25:24:1, pH 6.5–8) was added before processing as per manufacturer’s guidelines, with the exception of the bead-beating step, which was performed in 2 mL tubes on a Precellys 24 bead beater at 6500 RPM for 45 s. Negative controls consisting of 1X sterile phosphate-buffered saline were included with all extractions and processed in an identical manner to amniotic fluid samples.

### 2.6. 16S rRNA Gene Amplification and Barcoding

Extracted DNA was amplified using the PacBio uni-tagged 27F/1492R primers, as previously described [9]. PCR was carried out in 30 µL reactions consisting of 15 µL Accustart II PCR ToughMix (QuantaBio), 0.3 µM each of the forward and reverse primers, 0.75 µL each of the ArcticZymes PCR decontamination kit dsDNase and DTT (ArcticZymes), 6.6 µL nuclease-free water, and 6 µL of template or nuclease-free water (for the no-template controls). PCR was performed in a Veriti Thermal Cycler (Applied Biosystems) with the following thermocycling conditions: an initial heating step at 94 °C for 3 min; 40 cycles of 94 °C for 30 s, 55 °C for 30 s and 72 °C for 2 min; and a final extension step of 72 °C for 7 min. PCR products were visualised on a QIAXcel automated electrophoresis system using a DNA high-resolution gel cartridge (run parameters OM500) to confirm the presence and size of amplicons. Irrespective of whether a visible amplicon was present, all samples proceeded to barcoding and sequencing.

Barcoding was performed using the PacBio uni-tagged barcodes 1F-3F and 16R-30R. PCR products were purified using Agencourt AMPure XP magnetic beads (Beckman Coulter), normalised to 1 ng/µL, then used as template in a barcoding PCR, in order to generate asymmetrically barcoded amplicons. Secondary PCR was carried out in 25 µL reactions containing 12.5 µL AccuStart II ToughMix, 0.3 μM each of the forward and reverse primers, 3 μL of water and 2 μL of template. The PCR amplification program consisted of an initial heating step at 94 °C for 3 min; 10 cycles of 94 °C for 30 s, 55 °C for 30 s and 72 °C for 2 min; and a final extension step of 72 °C for 7 min.

### 2.7. PacBio Sequencing

Barcoded 16S rRNA gene amplicons were pooled in equimolar concentrations based on QIAXcel quantitation of the target band (~1.5 kB). The amplicon pool was then concentrated using Agencourt AMPure XP magnetic beads and eluted into 25 µL. The pool was subsequently gel purified from a 1.2% agarose gel through excising the band of the correct size and extracting using the QIAquick Gel Extraction kit (QIAGEN, Clayton, Victoria, Australia) as per manufacturer’s instructions. A total of 500 ng of purified DNA was used for library preparation at the Queensland University of Technology Genomics Research Centre. Here, SMRTbell adapters were ligated onto barcoded PCR products, and the libraries were sequenced on a PacBio Sequel system (version 5.1.0) on a single SMRT cell.

### 2.8. Sequence Processing

PacBio raw reads were demultiplexed to obtain circular consensus sequence (CCS) reads for each sample. CCS reads were filtered to retain only those with a minimum of three full passes and 99.9% sequence accuracy.

Demultiplexed sequence data was processed using mothur version 1.44.1 [29]. Fastq files were first converted to fasta files and merged. Filtering was performed to remove sequences of <1336 bp or >1743 bp and those containing homopolymers of >9 bases. Alignment was performed using the SILVA reference alignment database (v138). Chimeric sequences were identified and removed using vsearch. Sequences mapping to non-bacterial taxa were also removed. Sequences were clustered into OTUs with a similarity cut-off of 0.03 using the cluster.split command. Subsampling was performed to 867 reads based on a mean Good’s coverage of 0.989 (range 0.9–1.0). This excluded five samples from further analysis (four preterm, one term). Taxonomic identification of OTUs of interest was performed using BLAST with a cut-off of >99% sequence identity.

### 2.9. Identification of Contaminant Sequences

Due to the extremely-low-biomass nature of amniotic fluid, no computational removal of potential contaminant sequences was performed, in line with our previous reports on low-biomass microbiome environments [9,10,30,31,32]. Instead, a list of all OTU’s present in our negative extraction controls, and the associated read numbers and sequence taxonomies are provided (Table 1). Unfortunately, computational removal of potentially contaminating sequences using programs such as Decontam and MicroDecon from extremely-low-biomass samples such as amniotic fluid generally gives poor results, especially with low sample sizes [33], frequently resulting in removal of real sequences. Karstens et al. [34] demonstrated this in samples as low as 11 ng/µL, a sample biomass approximately 10 times higher than what is typically found in human amniotic fluid. There are a number of reasons why post hoc removal of potential contaminants is unreliable in extremely-low-biomass samples, but in the case of the popular decontamination pipeline Decontam, it is a result of the inability to accurately quantify microbial DNA in the sample. Amniotic fluid predominantly consists of human DNA, and as such, DNA input is not normalised to a uniform figure across all samples as would be typically seen in microbial profiling experiments from samples such as stool. Hence, reporting of sequences detected in negative controls is currently the most transparent and bias-free manner of controlling for reagent-based contamination in extremely-low-biomass samples.

### 2.10. Statistical Analysis

Demographic, pregnancy and birth characteristics were summarised according to timing of birth (term/preterm) using median and interquartile range for continuous data and frequency distributions for categorical data. Continuous outcomes were compared using Mann–Whitney tests and categorical outcomes were compared using Fishers Exact tests.

Cytokine concentrations for all births, term births and preterm births were summarised using median and interquartile ranges, with cytokine concentrations between term and preterm births contrasted using Mann–Whitney tests. Regression analyses (bacterial vs. cytokine profiles) were done at the genus level (as assigned by mothur) due large numbers of zero values at the OTU level making this analysis difficult.

Amniotic fluid richness (number of observed OTUs) and Shannon diversity were generated using mothur and compared between groups using a Mann–Whitney U-test. Bray–Curtis distances were generated with mothur and analysed via ANOSIM. Differences in the relative abundance of individual OTUs were assessed using metastats [35] in mothur, and only reported for OTUs which made up a mean relative abundance of >0.5% and which were present in >1 sample.

For linear regression modelling, each genus that was present in four or more samples was classified into present or absent and entered into the model as a binary variable. *Novosphingobium* sp. was present in all 31 women, so it was binarised into the 50% with the highest relative abundance and the 50% with the lowest relative abundance. Linear regression models with robust standard errors were used to assess associations between the presence or absence of each genus and levels of the cytokines IL-10, IL-1β, IL-6, TNF-α and MCP-1. The distribution of the cytokines was skewed so these outcomes were log transformed, with the exception of IL-1β, which was square-root transformed. Results are presented as back-transformed marginal means and 95% confidence intervals. Adjusted linear regression models were adjusted for sex and history of preterm birth.

SPSS version 25.0 (International Business Machines, Armonk, NY, USA), SAS Enterprise Guide version 7.1 (SAS institute Inc., Cary, NC, USA) and STATA version 16 (StataCorp LLC., College Station, TX, USA) statistical software were used for data analysis. For all tests, results were considered significant if *p* < 0.05.

## 3. Results

This was a sub-study nested within a larger cohort [28] in which 492 women were recruited, 52 of whom gave birth preterm (10.6%). The current study utilised samples from 39 women in the original cohort. This included all preterm births < 34 weeks’ GA (range 21–34 weeks; n = 17) and 22 term controls from uncomplicated pregnancies (range 38–42 weeks).

Of these, two samples produced no reads, and six were excluded via our subsampling method. This left 31 samples for inclusion in final analyses (final cohort: 11 preterm, 20 term). Of the three DNA extraction controls included, two produced reads (Table 1); both PCR controls produced no reads. All reads were uploaded onto the Sequence Read Archive under SubmissionID/BioProjectID: SUB11569063/PRJNA846514. Characteristics of the final cohort are presented in Table 2.

### 3.1. Amniotic Fluid Cytokines

Mid-trimester amniotic fluid samples from women who went on to deliver preterm had significantly higher levels of TNF-α compared to those collected from women who went on to deliver at term (*p* = 0.042) (Table 3, Figure 1). No significant differences were detected in IL-10, IL-1β, IL-6 or MCP-1 levels based on timing of birth.

### 3.2. No Difference in Amniotic Fluid Bacterial DNA Profiles from Term and Preterm Deliveries

Bacterial DNA profiles generated from amniotic fluid samples were low in diversity and richness (Figure 2). No differences were observed in the richness (*p =* 0.28) nor Shannon diversity (*p =* 0.17) of samples collected from women who delivered at term vs. those who delivered preterm. Further, dissimilarity analysis based on Bray–Curtis distances revealed no differences in the communities detected in these samples (ANOSIM *p* = 0.124).

The composition of the microbiome in these samples was relatively simple, with just 10 OTUs making up a mean relative abundance of >0.5% (Figure 3). Both term and preterm samples were dominated by an OTU which mapped to *Novosphingobium* sp. and could not confidently be classified to the species level; however, this OTU was also highly abundant in our negative controls, followed by OTU’s mapping to *Sphingomonas* sp. and the Family Sphingmonadaceae (Table 1). We were not able to identify any significant differences in the relative abundance of individual OTUs in these samples (all *p* > 0.23). As reads mapping to *Novosphingobium* sp., *Sphingomonas* sp. and the family Sphingmonadaceae are highly likely to represent DNA extraction kit contaminants, we have also included a figure with these dominant OTUs removed (Figure 4). This allows an easier visualisation of the likely true sample composition and makes it easier to discern between what was considered a positive and negative sample. Bacterial DNA unlikely to have originated from extraction controls was identified in 20/31 (64.5%) mid-gestation amniotic fluid samples. Nevertheless, some taxa of potential clinical interest were identified, including *Fusobacterium pseudoperiodonticum* (present in a single preterm sample at a relative abundance of 20.9%), *Bacillus licheniformis* (present in a single preterm sample at a relative abundance of 2.8%) and *Burkholderia cepacia* (present in a single term sample at a relative abundance of 50.6% and in two preterm samples at relative abundances of 11.3% and 1.04%).

### 3.3. Association between Amniotic Fluid Bacterial DNA Profiles and Cytokine Levels

Of the OTUs that yielded a relative abundance >0.5%, only four had enough non-zero observations to satisfy the requirements for a linear regression analysis with cytokine data. As a result, we conducted these analyses on the genera that were present in four or more samples, as per the original mothur analysis. In the adjusted analysis, IL-1β levels were significantly higher in samples containing reads mapping to *Sphingomonas* sp. (back-transformed marginal mean (95% CI) absent/present: 0.60 (0.39, 0.87)/1.54 (0.73, 2.65) pg/mL; *p =* 0.031). TNF-α levels were also significantly higher in samples containing reads mapping to *Sphingomonas* sp. (marginal geometric mean (95% CI) absent/present: 3.87 (3.00, 4.99)/7.26 (4.35, 12.11) pg/mL; *p* = 0.037), whilst IL-6 levels were significantly increased in samples with reads that mapped to *Pelomonas* sp. (marginal geometric mean (95% CI) absent/present: 79.10 (45.30, 138.14)/469.40 (163.70, 1345.96) pg/mL; *p* = 0.005) (Table 4). Results were similar for unadjusted analyses (Appendix A). The associations for *Sphingomonas* sp. should be interpreted with caution, as this genus was also identified in high numbers in our negative extraction controls (Table 1).

## 4. Discussion

Differences in amniotic fluid bacterial DNA profiles from pregnancies that ended in term or preterm birth could not be detected in this cohort. These data support previous findings in a Swedish cohort, in which bacterial DNA presence and bacterial DNA profiles were not associated with birth outcomes [10]. By contrast, amniotic fluid samples collected late in pregnancy (26–35 weeks gestation) have been shown to be predictive of chorioamnionitis stage in preterm deliveries [6], potentially suggesting that MIAC cannot be detected in amniotic fluid samples taken early in pregnancy. In the present study, samples were collected at 15–20 weeks gestation, and the earliest preterm birth in the cohort was at 21 weeks gestation (median 33 weeks). It is possible that this lag in time between sample collection and onset of preterm birth may account for the lack of difference in the sample profiles. Further, our study was limited in sample size, particularly in relation to spontaneous preterm births (six cases), and therefore may have lacked the statistical power to detect differences in early amniotic fluid bacterial DNA profiles.

While no statistical differences could be identified between term and preterm samples, DNA associated with potential pathogens could be identified in samples from women who went on to deliver preterm. In one preterm sample, *F. pseudoperiodonticum* was identified in high abundance (20.9% relative abundance). *F. pseudoperiodonticum* is a recently identified bacterium recovered from the human oral cavity [36]. Other species of oral *Fusobacterium* sp. have been repeatedly associated with preterm birth [11]. Indeed, rodent studies have demonstrated that oral *Fusobacterium nucleatum* is able to access the intra-amniotic space via a haematogenous route and trigger preterm birth and still birth [15,37]. The outcome of the pregnancy from which this OTU was identified was preterm delivery of a live infant at 30 weeks’ gestation, followed by perinatal mortality. Another potential pathogen detected in the amniotic fluid of a woman who went on to deliver preterm was *Bacillus licheniformis* (relative abundance of 2.8%). While this taxa has not previously been identified in amniotic fluid, other species of *Bacillus* sp. have been identified in the human placenta, leading to severe infection and foetal demise [38]. The pregnancy in the present cohort from which *B. licheniformis* was identified ended in preterm still birth at 21 weeks gestation. *Burkholderia cepacia*, a potential pathogen, was identified in one term sample (relative abundance 50.6%) and two preterm samples (relative abundances of 11.3% and 1.04%). This species has previously been reported as a cause of neonatal sepsis in a full term infant [39], while other species of the genus have been reported to cause intra-amniotic infection and preterm labour [40,41]. The results of our study therefore highlight the potential to detect preterm-birth-associated pathogens in amniotic fluid samples taken early in pregnancy.

In addition to potential pathogens, within the preterm birth group, we also identified an OTU which mapped to *L. mindensis*. To our knowledge, this species has not previously been recovered from the female genital tract; however, it is part of a group of *Lactobacillus* species that cannot easily be separated by their 16S rRNA gene sequence [42]. Other species of this genus are typical of the vaginal microbiome [43]. The pregnancy from which this OTU was identified ended in preterm pre-labour rupture of membranes (PPROM) and preterm delivery of a live infant at 31 weeks’ gestation. The presence of this OTU in the amniotic fluid of a woman who went on to deliver preterm may have been an early sign of cervical insufficiency.

The majority of the bacterial DNA reads recovered from these samples mapped to *Novosphingobium* sp., with a large number of reads also mapping to *Sphingomonas leidyi*. These taxa are likely to be contaminants in our workflow, as they were also detected in our negative extraction controls (Table 1). Other commonly identified taxa in this cohort include *Ralstonia pickettii*, *Acidovorax wautersii* and *Pelomonas puraquae*. *R. pickettii* has previously been identified in Australian amniotic fluid samples taken at caesarean section [9], while other species of *Ralstonia* sp. have been described as bona fide residents of the basal plate of the placenta [44]. *A. wautersii* has previously been isolated from human blood [45], while other species of this genus have been isolated from mid-trimester amniotic fluid [10]. *P. puraquae* has been detected in amniotic fluid [9,10] and meconium [9]. These taxa may therefore represent “normal” amniotic fluid species. However, it is important to note that the presence of DNA from these taxa within the intra-amniotic space does not necessarily indicate the viability of these species.

To the best of our knowledge, our study is the first to examine amniotic fluid cytokine levels alongside matched bacterial DNA profiles. Levels of IL-1β and TNF-α were increased in samples containing reads mapping to *Sphingomonas* sp., whilst IL-6 levels were higher in samples with reads that mapped to *Pelomonas* sp. Taken together, this may be indicative of foetal exposure to inflammation arising from the presence of bacterial agonists (DNA or non-viable/viable cells); however, the result must be interpreted with caution considering the relatively high abundance of *Sphingomonas* sp. reads detected in our negative control samples, combined with the fact that previous studies have also reported the presence of these and other cytokines in the absence of detectable bacterial DNA, so-called ‘sterile inflammation’ [18]. We have, however, previously documented an association between *Pelomonas puraquae* in late gestation amniotic fluid and elevated levels of another cytokine, granulocyte colony-stimulating factor (G-CSF) [9], which was not examined in the present study. Although we observed elevated TNF-α levels in amniotic fluid from preterm pregnancies compared to those who delivered at term, the levels documented in both groups were very low (at least 10-fold lower) compared to previous studies reporting intra-amniotic inflammation [28,46] and, thus, are unlikely to be of biological relevance.

In this cohort, mid-trimester amniotic fluid bacterial DNA profiles were not predictive of preterm birth. While samples taken from pregnancies that ended in preterm delivery showed elevated TNF-α levels, amniotic fluid cytokine levels were overall not predictive of preterm birth. The results of this small study therefore indicate that mid-trimester sampling is not useful for identifying women who are at risk of delivering preterm. However, in a small number of cases, potential perinatal pathogens may be identified in early amniotic fluid samples.

## Figures and Tables

**Figure 1 microorganisms-11-01698-f001:**
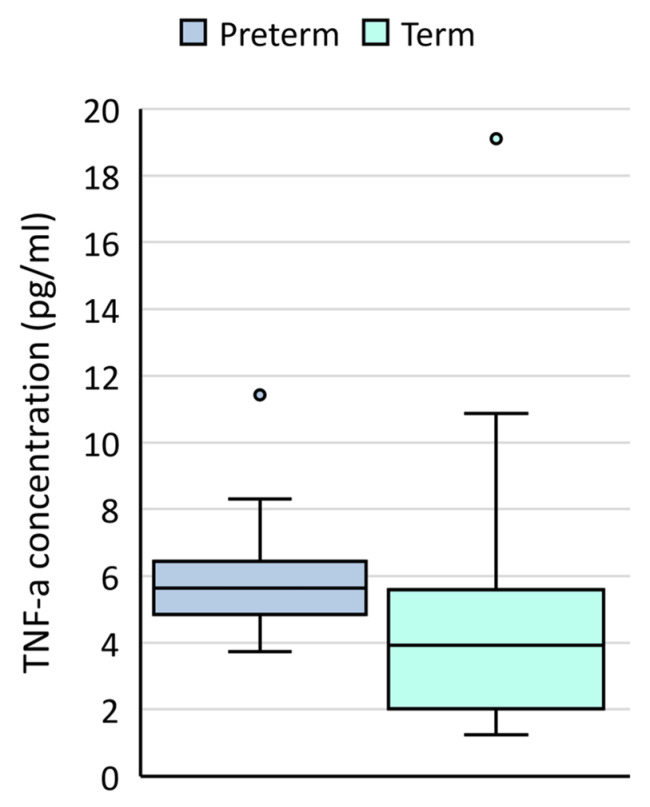
TNF-α concentrations (pg/mL) in mid-trimester amniotic fluid samples collected from women who went on to deliver at term (light blue boxes) or preterm (dark blue boxes).

**Figure 2 microorganisms-11-01698-f002:**
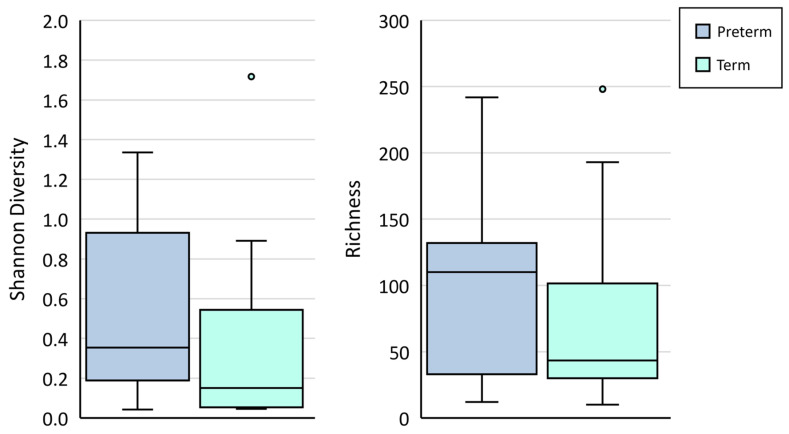
Shannon diversity and richness (number of OTUs) in mid-trimester amniotic fluid samples collected from women who went on to deliver at term (light blue boxes) or preterm (dark blue boxes).

**Figure 3 microorganisms-11-01698-f003:**
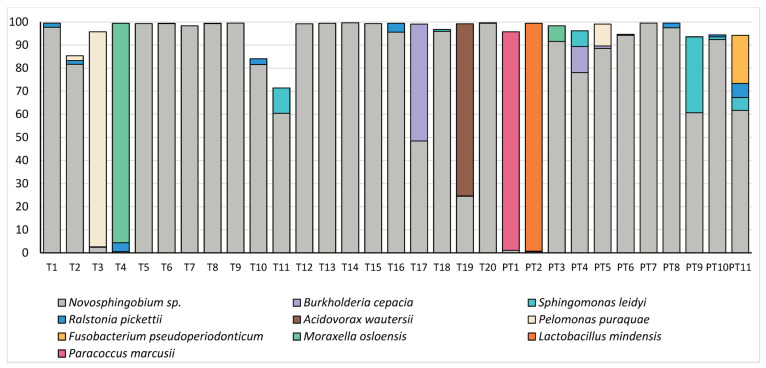
Relative abundances (%) of taxa which made up >0.5% of amniotic fluid bacterial DNA profiles in term (T) and preterm (PT) samples.

**Figure 4 microorganisms-11-01698-f004:**
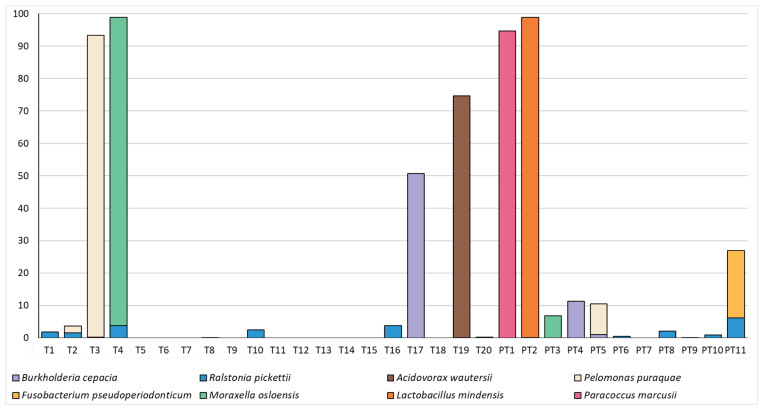
Relative abundances (%) of taxa which made up >0.5% of amniotic fluid bacterial DNA profiles in term (T) and preterm (PT) samples and were not present in extraction controls in high numbers.

**Table 1 microorganisms-11-01698-t001:** Contaminating genera detected in our negative extraction and PCR controls. Data are number of reads.

	Extraction Controls	PCR Controls
*Novosphingobium* sp.	13,999	0
*Sphingomonas* sp.	570	0
Unclassified Sphingomonadaceae	213	0
*Ralstonia* sp.	73	0
*Rhizorhapis* sp.	5	0
*Moraxella* sp.	1	0
*Pelomonas* sp.	1	0
*Parablastomonas* sp.	1	0

**Table 2 microorganisms-11-01698-t002:** Demographic and birth characteristics of women in the study by timing of birth (preterm/term). Data are median (IQR) or n (%).

	Preterm	Term	*p*-Value
N	11	20	
Maternal age (years)	36.4 (29.1, 40.6)	33.0 (25.4, 36.6)	0.301
Ethnicity			0.458
Caucasian	7 (63.6)	9 (45.0)	
Other *	4 (36.4)	11 (55.0)	
Nulliparous	3 (27.3)	9 (45.0)	0.452
Previous preterm birth	3 (27.3)	0 (0.0)	**0.037**
Smoking during pregnancy	2 (18.2)	2 (10.0)	0.602
Preterm pre-labour rupture of membranes	6 (54.5)	0 (0.0)	**0.001**
Threatened preterm labour	5 (45.5)	0 (0.0)	**0.003**
Clinical chorioamnionitis	1 (9.1)	0 (0.0)	0.355
Spontaneous labour	4 (36.4)	14 (70.0)	0.128
Spontaneous onset ^	6 (54.5)	14 (70.0)	**<0.001**
Gestational age at birth	33.0 (27.0, 34.0)	39.0 (39.0, 40.0)	**<0.001**
Birthweight (g)	1836 (1148, 2180)	3507.5 (3132.5, 3915.0)	**<0.001**
Male	6 (54.5)	11 (55.0)	1.000
Stillborn	3 (27.3)	0 (0.0)	0.037

* includes one case of unknown ethnicity. ^ Spontaneous onset includes anyone who had a spontaneous labour or pre-labour rupture of membranes.

**Table 3 microorganisms-11-01698-t003:** Distribution of cytokines by timing of birth. Data are pg/mL, median (IQR).

	All	Preterm	Term	*p*-Value
**IL-10**	5.3 (2.5, 11)	5.9 (5.1, 12.5)	3.9 (2.0, 9.0)	0.097
**IL-1β**	0.8 (0.3, 1.1)	1.0 (0.8, 1.7)	0.5 (0.2, 0.9)	0.058
**IL-6**	87.1 (33.3, 377.9)	131.6 (29.6, 409.2)	69.9 (35.0, 319.7)	0.648
**MCP-1**	925.1 (685.6, 1161.4)	904.0 (679.2, 1132.5)	941.9 (756.7, 1165.0)	0.587
**TNF-α**	4.9 (3.4, 6.4)	5.6 (4.9, 6.4)	3.9 (2.2, 5.4)	**0.042**

**Table 4 microorganisms-11-01698-t004:** Adjusted ^#^ mean cytokine levels and their 95% confidence intervals, by presence of each genus.

		IL-10		IL-1β		IL-6		MCP-1		TNF-α	
		Marginal Geometric Mean	*p*	Back-Transformed Marginal Mean	*p*	Marginal Geometric Mean	*p*	Marginal Geometric Mean	*p*	Marginal Geometric Mean	*p*
***Novosphingobium* sp. ***	**Low**	5.85 (3.56, 9.60)	0.483	0.81 (0.45, 1.28)	0.903	140.37 (59.32, 332.14)	0.229	967.97 (754.20, 1242.34)	0.386	5.06 (3.92, 6.53)	0.318
	**High**	4.39 (2.33, 8.31)		0.77 (0.37, 1.31)		71.67 (35.94, 142.93)		848.09 (712.96, 1008.84)		3.96 (2.64, 5.94)	
**Sphingomonadaceae_** **Unclassified**	**Absent**	6.75 (4.08, 11.16)	0.178	0.64 (0.41, 0.94)	0.245	122.24 (54.90, 272.18)	0.549	795.90 (666.84, 949.95)	0.105	4.76 (3.65, 6.21)	0.621
	**Present**	3.71 (1.90, 7.23)		0.99 (0.49, 1.66)		82.31 (32.33, 209.51)		1031.44 (805.96, 1320.00)		4.16 (2.66, 6.51)	
***Ralstonia* sp.**	**Absent**	4.80 (2.52, 9.12)	0.758	0.71 (0.36, 1.19)	0.556	100.16 (45.27, 221.61)	0.995	860.88 (706.77, 1048.60)	0.460	4.37 (3.01, 6.34)	0.831
	**Present**	5.42 (3.51, 8.36)		0.89 (0.51, 1.38)		100.49 (44.12, 228.87)		968.71 (757.42, 1238.95)		4.59 (3.46, 6.09)	
***Sphingomonas* sp.**	**Absent**	4.29 (2.63, 7.01)	0.139	0.60 (0.39, 0.87)	**0.031**	89.84 (45.21, 178.51)	0.491	862.02 (750.95, 989.52)	0.363	3.87 (3.00, 4.99)	**0.037**
	**Present**	8.81 (4.17, 18.61)		1.54 (0.73, 2.65)		144.05 (47.23, 439.34)		1067.18 (678.99, 1677.31)		7.26 (4.35, 12.11)	
***Pelomonas* sp.**	**Absent**	4.53 (2.95, 6.96)	0.113	0.85 (0.54, 1.22)	0.168	79.10 (45.30, 138.14)	**0.005**	895.21 (754.59, 1062.04)	0.410	4.50 (3.39, 5.97)	0.734
	**Present**	10.43 (4.12, 26.36)		0.46 (0.15, 0.95)		469.40 (163.70, 1345.96)		979.78 (834.87, 1149.85)		4.21 (3.37, 5.25)	

^#^ Adjusted for sex, and history of preterm birth. * For *Novosphingobium* sp., values lower than the median (median = 0.943) were classified as low relative abundance, and values equal to or greater than the median were classified as high relative abundance.

## Data Availability

All sequence reads generated by this study are publicly available on the Sequence Read Archive under SubmissionID/BioProjectID: SUB11569063/PRJNA846514.

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
