# Peer review of "Characterisation of Mid-Gestation Amniotic Fluid Cytokine and Bacterial DNA Profiles in Relation to Pregnancy Outcome in a Small Australian Cohort"

_microorganisms, 2023, doi:10.3390/microorganisms11071698_

Round 1

Reviewer 1 Report

This study explored the association exists between bacterial and cytokine profiles in Australian mid-gestation amniotic fluid samples from preterm and term births. The topic is interesting, while some data needs to be added.

1.    Amniotic fluid bacterial phylum and genus levels

2.    Metabolites in the amniotic fluid.

3.    The association exists between bacterial and metabolites.

Author Response

We would like to thank the reviewer for taking the time to review our manuscript. Please find below our responses to the issues raised.

This study explored the association exists between bacterial and cytokine profiles in Australian mid-gestation amniotic fluid samples from preterm and term births. The topic is interesting, while some data needs to be added.

  1. Amniotic fluid bacterial phylum and genus levels

It is unclear how analysing our data at the phylum and genus level would benefit the manuscript. We deliberately selected PacBio HiFi sequencing of the full-length 16S rRNA gene to enable species-level taxonomy for the majority of cases; this allows more clinically relevant associations to be explored. With this in mind, we feel that inclusion of phylum and genus-level associations would be redundant data and if anything may be considered ‘p-hacking’ as it could be conceived as working backwards to try and find statistical significance for one or more taxa.

  1. Metabolites in the amniotic fluid.
  2. The association exists between bacterial and metabolites.

While we agree that data on bacterial and host metabolites present in the amniotic fluid in relation to the amniotic fluid microbiome and pregnancy outcome would be of much clinical interest, it is unfortunately not possible for two reasons. First, the samples described in this study were exhausted during the bacterial DNA and host RNA analyses described, and even if there were aliquots remaining, the samples had not been collected in a manner suitable for analysis of labile metabolites. We will, however, certainly keep this suggestion in mind if the opportunity arises to examine amniotic fluid samples in this context again in the future.

Reviewer 2 Report

The manuscript:  Characterisation of mid-gestation amniotic fluid cytokine and bacterial DNA profiles in relation to pregnancy outcome in a small Australian cohort

 Lisa F. Stinson, Yey Berman, Shaofu Li, Jeffrey A. Keelan, Jan E. Dickinson, Dorota A. Doherty, John P. Newnham and Matthew S. Payne

Title: The title of the manuscript is appropriate

Introduction: enough; represent the essence of the problem; does not require change

The study is relevant for the scientific community, since the problem of preterm birth is very acute. It is chorioamnionitis that can cause premature birth and spontaneous miscarriages in the second trimester of pregnancy.

Methodology: meets the requirements of the journal and the branch of knowledge; does not require modification;

The authors obtained samples of amniotic fluid at 15-20 weeks of gestation. It is very difficult, there must be strict indications for amniocentesis. These were genetic indications.

In this regard, I have a question: what are the results of genetic studies in these women. How many women were found to have aneuploidies and other genetic defects? Was there a link between genetic damage and the presence of pro-inflammatory or anti-inflammatory cytokines in the amniotic fluid?

The authors used complex and reliable diagnostic methods

Statistics: Adequate methods of statistical data processing were used

Results:

Lines 239-242 - the phrase "The current study utilised samples from 39 women in the original cohort, including all preterm births <34 weeks' GA (range 21-34 weeks), combined with term controls from uncomplicated pregnancies (17 preterm, 22 term)" is not clear. 17 women with preterm labor and 22 with term labor? What was the gestational age in both groups? It must be specific.

Discussion of the results: sufficient and consistent with the main results obtained.

Literature: sufficient for the article and does not require processing

Article design: meets the requirements of the journal

Conclusion: the article meets the requirements of the journal and can be published after a little revision.

Author Response

We would like to thank the reviewer for taking the time to review our manuscript. Please find below our responses to the issues raised.

The authors obtained samples of amniotic fluid at 15-20 weeks of gestation. It is very difficult, there must be strict indications for amniocentesis. These were genetic indications.

In this regard, I have a question: what are the results of genetic studies in these women. How many women were found to have aneuploidies and other genetic defects? Was there a link between genetic damage and the presence of pro-inflammatory or anti-inflammatory cytokines in the amniotic fluid?

These are great questions, but unfortunately they are not ones we are able to answer. Although the amniotic fluid samples were obtained during routine amniocenteses, we were only granted ethical permission to access the sample for analyses in line with our microbial and inflammatory hypotheses relative to pregnancy outcome. We were not permitted to access data from the results of genetic aneuploidy screening that each participant was having conducted. Considering the original samples were collected over 10 years ago now, trying to obtain these data in retrospect would be very difficult, if not impossible.

Results:

Lines 239-242 - the phrase "The current study utilised samples from 39 women in the original cohort, including all preterm births <34 weeks' GA (range 21-34 weeks), combined with term controls from uncomplicated pregnancies (17 preterm, 22 term)" is not clear. 17 women with preterm labor and 22 with term labor? What was the gestational age in both groups? It must be specific.

We have changed the text at lines 217-220 to make the grouping clearer and also added the gestational age range for the term samples: “The current study utilised samples from 39 women in the original cohort. This included all preterm births <34 weeks’ GA (range 21-34 weeks; n=17), and 22 term controls from uncomplicated pregnancies (range 38-42 weeks).”

Round 2

Reviewer 1 Report

Accept in its current form.